# Maternally inherited genetic variants of *CADPS2* are present in Autism Spectrum Disorders and Intellectual Disability patients

Elena Bonora[1,†], Claudio Graziano[1,†], Fiorella Minopoli[1,2], Elena Bacchelli[2], Pamela Magini[1], Chiara Diquigiovanni[1], Silvia Lomartire[2], Francesca Bianco[1], Manuela Vargiolu[1], Piero Parchi[3], Elena Marasco[4], Vilma Mantovani[1,4], Luca Rampoldi[5], Matteo Trudu[5], Antonia Parmeggiani[3], Agatino Battaglia[6], Luigi Mazzone[7], Giada Tortora[1], IMGSAC[8,‡], Elena Maestrini[2], Marco Seri[1,*] & Giovanni Romeo[1]

## Abstract

Intellectual disability (ID) and autism spectrum disorders (ASDs) are complex neuropsychiatric conditions, with overlapping clinical boundaries in many patients. We identified a novel intragenic deletion of maternal origin in two siblings with mild ID and epilepsy in the *CADPS2* gene, encoding for a synaptic protein involved in neurotrophin release and interaction with dopamine receptor type 2 (D2DR). Mutation screening of 223 additional patients (187 with ASD and 36 with ID) identified a missense change of maternal origin disrupting CADPS2/D2DR interaction. *CADPS2* allelic expression was tested in blood and different adult human brain regions, revealing that the gene was monoallelically expressed in blood and amygdala, and the expressed allele was the one of maternal origin. *Cadps2* gene expression performed in mice at different developmental stages was biallelic in the postnatal and adult stages; however, a monoallelic (maternal) expression was detected in the embryonal stage, suggesting that *CADPS2* is subjected to tissue- and temporal-specific regulation in human and mice. We suggest that *CADPS2* variants may contribute to ID/ASD development, possibly through a parent-of-origin effect.

**Keywords** autism spectrum disorders; CADPS2; intellectual disability; monoallelic expression; mutation screening

**Subject Categories** Genetics, Gene Therapy & Genetic Disease; Neuroscience

## Introduction

Intellectual disability (ID) is a neurodevelopmental disorder characterized by a below-average score on tests of mental ability and limitations in daily life functions, with a frequency of 1–3% (van Bokhoven, 2011). Autism spectrum disorders (ASDs) are characterized by impaired social interactions and communication and stereotyped behaviors with onset before 3 years of age. ASDs are currently estimated to affect ~1% of children (Autism and Developmental Disabilities Monitoring Network 2009) and significantly skewed toward boys, with a sex ratio of 4:1 (Fombonne, 2005).

Although many monogenic and chromosomal causes of ID are known, a "multiple hit"/oligogenic model is emerging, where combinations of variants are necessary to disrupt normal neuronal development and underlie a range of disorders from idiopathic epilepsy to autism and ID (Coe *et al*, 2012). Many genes involved in synaptic function were shown to play a role in neurodevelopmental disorders (Gilman *et al*, 2011). Nevertheless, although definite epidemiological data are lacking, causative mutations remain unknown in the majority of ID/ASD patients.

We have identified a novel intragenic deletion and a potentially deleterious rare single nucleotide variant (SNV) in the Ca$^{2+}$-dependent activator protein for secretion 2 (*CADPS2*) gene, leading to the disruption of its interaction with the dopamine receptor type 2 (D2DR; Binda *et al*, 2005), in individuals with either ASD/ID. *CADPS2* is an excellent candidate for neurologic development abnormalities, given that it is predominantly expressed in the nervous system, and is involved in neurotrophin-3 (NT-3) and

1 Unit of Medical Genetics, Department of Medical and Surgical Sciences, S. Orsola-Malpighi Hospital, University of Bologna, Bologna, Italy
2 Department of Pharmacy and Biotechnology, University of Bologna, Bologna, Italy
3 Department of Neurology, University of Bologna, Bologna, Italy
4 CRBA, S. Orsola-Malpighi Hospital, Bologna, Italy
5 Molecular Genetics of Renal Disorders Unit, Division of Genetics and Cell Biology, San Raffaele Scientific Institute, Milan, Italy
6 Stella Maris Clinical Research Institute for Child and Adolescent Neurology and Psychiatry, Calambrone (Pisa), Italy
7 Unit of Child Neuropsychiatry, IRCCS Ospedale Pediatrico Bambino Gesù, Roma, Italy
8 IMGSAC Institute of Neuroscience and Health and Society,, Newcastle University, Newcastle upon Tyne, Tyne and Wear, UK
  *Corresponding author. Tel: +39 51 2088421; Fax: +39 51 2088416; E-mail: marco.seri@unibo.it.
  †These authors share a joint first co-authorship.
  ‡A full list IMGSAC Consortium members can be found in the supplementary data.

brain-derived neurotrophic factor (BDNF) release (Sadakata & Furuichi, 2009, 2010). *CADPS2* maps to the "autism susceptibility locus 1" on chromosome 7q31-q33(Lamb *et al*, 2005) and is one of the genes that were shown to be downregulated in brains of autistic individuals (Voineagu *et al*, 2011).

The new functional SNV (p. Asp1113Asn) is of maternal origin, and also the novel intragenic deletion is maternally inherited. We show for the first time that *CADPS2* is maternally expressed in human blood and amygdala. We identified a cluster of differentially methylated CpG regions in the first intron of the gene, in bisulfite-treated DNA from the blood and amygdala. The differential methylation in this region is not observed in other brain areas, such as the cerebellum, where we found a biallelic expression of the gene. However, we could not identify a reproducible parent-of-origin methylation profile for these CpGs, suggesting that other genomic regions in CADPS2 or other regulatory mechanisms may be related to the presence of a monoallelic pattern of expression.

In adult mice, we observed a biallelic expression of the gene in all the postnatal and adult stages analyzed; however, we found a monoallelic maternal expression in the cerebellum at the embryonal stage E17.5. Therefore, we suggest that *CADPS2* is subjected to a fine temporal- and tissue-specific regulation, leading to a maternally expressed gene, where inherited mutations may contribute to the ID/ASD phenotype.

## Results

### Identification of a novel intragenic deletion in CADPS2

We ascertained a pair of siblings (male and female) with behavioral problems, borderline ID, and epilepsy. Array-CGH analysis detected an intragenic deletion of ~285 kb in *CADPS2* on chromosome 7q31.32 in both siblings, likely to be inherited from the deceased mother, since the father did not carry it (Fig 1A and B, and Supplementary Fig S1A).

The deletion was confirmed to be of maternal origin, by microsatellite markers and single nucleotide polymorphisms (SNPs) analysis, as shown in Fig 1B.

No deletions overlapping *CADPS2* are present in the Database of Genomic Variants (DGV), and this CNV was not present in ISCA (https://www.iscaconsortium.org/index.php) and Troina (http://gvarianti.homelinux.net/gvariantib37/index.php) databases. Only a small duplication overlapping exon 1 has been reported (Shaikh *et al*, 2009).

Two additional CNVs were identified in the affected siblings, but since they were already reported in DGV (Database of Genomic Variants) and defined non-pathogenic, we did not further characterize them (Supplementary Table S1).

We mapped the deletion boundaries between intron 3 and intron 28 of *CADPS2* via quantitative PCR (Fig 1C) and defined by long-range PCR the deletion breakpoint between bp 121,984,852–122,270,267 of chromosome 7 (hg19) (Fig 1D). Sequencing of *CADPS2*-coding exons in the two siblings excluded the presence of additional variants. Interestingly, we could not detect any *CADPS2* transcripts by RT–PCR, neither the wild-type nor in the aberrant one, in the blood of the two affected siblings (Supplementary Fig S1B and C).

### CADPS2 mutation screening in ASD/ID patients

Thirty-six Italian patients with ID and 187 probands with ASD [of which 94 from Italy and 93 from the International Molecular Genetic Study of Autism Consortium (IMGSAC) collection (IMGSAC, 2001)] were recruited for mutation screening of all exon and exon–intron boundaries of the *CADPS2* gene (NM_017954.10). Clinical characteristics of Italian ASD and ID individuals are reported in Supplementary Tables S1A and B, and clinical characteristics and inclusion criteria of IMGSAC ASD individuals have been previously reported (IMGSAC, 2001). Two synonymous (p. Ala26= and p. Ala402=) and five missense (p. Met630Thr, p. Phe645Val, p. Asp1088Asn, p. Asp1113Asn, and p.Val1137Met) heterozygous rare variants were identified (Table 1). Analysis of parental DNA indicated that all these variants were inherited (Fig 1E). Four missense changes were present in public databases (dbSNP, EVS), and their frequencies were not statistically different from the ones identified in our ASD/ID patients. In addition, the paternally inherited change p. Asp1088Asn did not co-segregate with the disease phenotype in the two ASD multiplex families where it was identified (Fig 1E). Three variants were novel: p. Ala402=, p. Ala26=, p. Asp1113Asn, and their frequency in the AD/ID group was statistically different form the control group (*P*-value = 0.0467; Fisher's *t*-test; 500 Italian chromosomes and EVS database). For the silent variants, no functional effect at the transcript level could be detected by RT–PCR or predicted by informatic tools such as ESE finder v3.0 and Human Splicing Finder (HSF) v2.4.1; hence, their functional effect was not tested further. Therefore, we focused on the p. Asp1113Asn missense variant for further functional studies, because this was the only one absent in all the databases and in the Italian controls and predicted damaging by PolyPhen-2 (PolyPhen score = 0.936; Table 1; Adzhubei *et al*, 2013).

### The novel change p.Asp1113Asn in CADPS2 disrupts the interaction with dopamine receptor type 2

We assessed the functional effect of the novel variant p.Asp1113Asn in the binding to dopamine receptor type 2 (D2DR), one of the few known interactors of CADPS2 (Binda *et al*, 2005), compared to wild-type CADPS2. After transfecting the different *CADPS2* cDNA clones in frame with a C-terminal V5 tag into the SHSY5Y cell line, derived from neuroblastoma and expressing dopaminergic markers (Sabens & Mieyal, 2010), co-immunoprecipitation assays of the corresponding cell lysates with an anti-D2DR antibody and Western blot analysis against the V5 tag showed that the novel missense change p. Asp1113Asn decreased the protein–protein interaction between CADPS2 and D2DR (Fig 1F). Co-immunoprecipitation with anti-V5 antibody and western blotting analysis against D2DR confirmed that the missense change interfered with CADPS2 and D2DR binding (Supplementary Fig S2).

### CADPS2 allelic expression analysis reveals that the gene is maternally expressed in blood

We noticed that the majority of the missense variants that were absent in Italian controls and co-segregated with the phenotype were of maternal origin, including the novel p. Asp1113Asn functional change (Fig 1F). Furthermore, the novel intragenic *CADPS2*

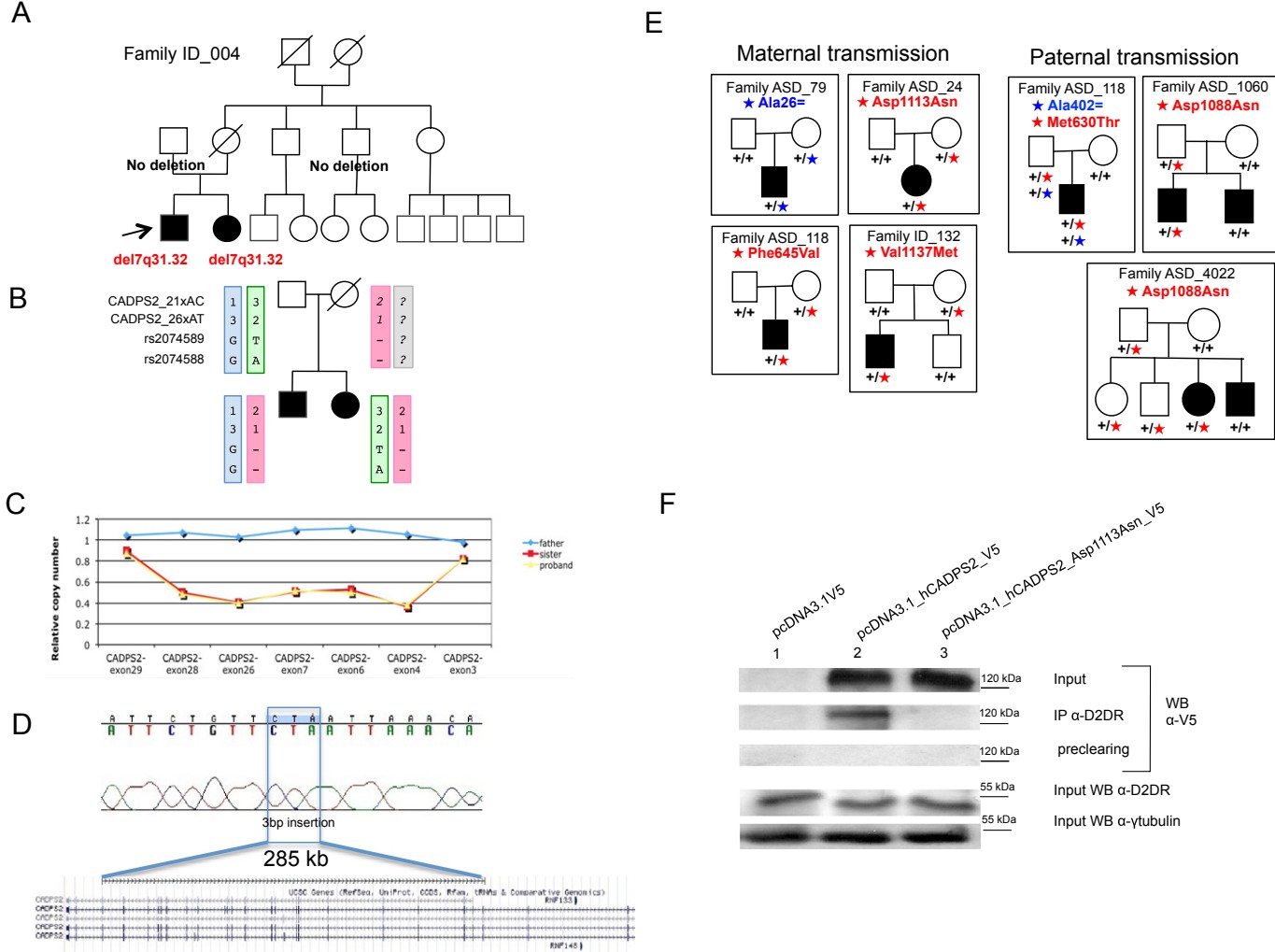

**Figure 1.** ***CADPS2* deletion mapping and parental origin of *CADPS2* coding variants.**

A   Family tree of the ID sibs carrying the novel intragenic deletion.

B   Microsatellite and SNP analysis showing the maternal inheritance of *CADPS2* intragenic deletion.

C   Fine mapping of *CADPS2* deletion by real-time qPCR using different probes across the region. All data were normalized using as reference gene *FOXP2*.

D   *CADPS2* deletion breakpoint mapping, showing the insertion of 3 base pairs at the breakpoint (upper panel) and the corresponding location on chromosome 7q (UCSC Genome Browser).

E   Parental origin of the coding *CADPS2* variants identified in the mutation screening. In blue are indicated the synonymous changes, and in red the missense changes. It is worth noting that in the family with more than one sib, the maternal variant p.Val1137Met was transmitted to the affected son (ID case) and not to the unaffected brother.

F   Analysis of the effect of CADPS2 variant p. Asp1113Asn in D2DR binding, in SHSY5Y cells transfected with the different *CADPS2* constructs tagged with V5 epitope tag: Immunoprecipitation was performed with a rabbit anti-D2DR antibody; western blotting was performed with a mouse anti-V5 tag antibody (first three panels) and with a rabbit anti-D2DR antibody; lowest panel: western blot for γ-tubulin on total cell lysates, as internal control of protein quantity. Second panel, lane 3: the co-immunoprecipitated CADPS2_Asp1113Asn_V5 is clearly diminished compared to wild-type (lane 2); the immunoblots are representative of three independent experiments.

deletion is of maternal origin, as shown in Fig 1B. Thus, we decided to test whether *CADPS2* might be subjected to a parent-of-origin regulation.

We first investigated the allelic expression of two *CADPS2* coding SNPs rs2251761 (exon 3) and rs2074589 (exon 17) in informative heterozygous controls of whom blood RNA was available. As shown in Fig 2A, in heterozygous individuals, only one allele was expressed in blood cDNA, both for rs2251761 (3 independent

controls tested) and for rs2074589 (4 independent controls tested). In order to establish whether the expressed allele was of maternal origin, we analyzed three different families for whom genomic DNA and blood mRNA of heterozygous individuals were available. One multigenerational family of controls was informative for SNP rs2251761. Segregation and expression analysis of the SNP showed that the *CADPS2* expressed allele in III-1 was of maternal origin (Fig 2B). In the family with the synonymous variant p. Ala402= the

**Table 1. Coding variants in *CADPS2* identified either in ASD or in ID patients.**

| Position on chr7 (hg19) | Type of change (NP_060424.9) NM_017954.10 | PolyPhen-2 score (HumDiv) | SIFT prediction (cutoff =0.05) | Parental origin | Het[a] in ASD/ID (N=223) | Het[a] in Italian controls (N=250) | Het[a] in EVS (European-American) | *P* value[b] (Fisher's *t*-test) |
|---|---|---|---|---|---|---|---|---|
| **g.122,5[26],314G>A none[c]** | p. Ala26= | na | na | Maternal | 1/223 (ASD) | 0 | 0 | **0.0467** |
| **g.122,255,252G>C none[c]** | p. Ala402= | na | na | Paternal[d] | 1/223 (ASD) | 0 | 0 | **0.0467** |
| g.122,114,544A>Grs199713510[c] | p. Met630Thr | 0.917 (possibly damaging) | 0 (damaging) | Paternal[d] | 1/223 (ASD) | 2/250 | 10/4113 | 0.477 |
| g.122,114,500A>C rs201536376[c] | p. Phe645Val | 0.001 (benign) | 0.531 (tolerated) | Maternal | 1/223 (ASD) | 0 | 9/4128 | 0.392 |
| g.122027130C>T rs76528953[c] | p. Asp1088Asn | 1 (probably damaging) | 0.001 (damaging) | Paternal | 2/223[e] (ASD) | 0 | 28/4145 | 0.429 |
| **g.122,019,472C>T none[c]** | p. Asp1113Asn | 0.936 (probably damaging) | 0.039 (tolerated) | Maternal | 1/223 (ASD) | 0 | 0 | **0.0467** |
| g.122,001,046C>T rs200984050[c] | p. Val1137Met | 0.997 (probably damaging) | 0.003 (damaging) | Maternal | 1/223 (ID) | 0 | 6/4122 | 0.294 |

[a]Het = number of heterozygous individuals.
[b]Fisher's exact test calculated considering the number of heterozygous individuals in the total control group [Italian and European-American from EVS (http://evs.gs.washington.edu/EVS/ accession March 2013].
[c]dbSNP entry reported for the corresponding variant; none = not present in dbSNP.
[d]The two paternal changes were present in the same ASD individual.
[e]The two individuals from multiplex ASD families inherited the change from the father; however in both cases, it did not co-segregate with the phenotype (see Fig 1F).
SNVs not found in dbSNP and EVS are shown in bold.

heterozygous father expressed only the C allele in blood cDNA, while the affected child expressed only the maternal wild-type G allele (Fig 2C). In the family carrying the p.Val1137Met variant, the expressed allele in the affected ID individual was the maternally inherited A allele, whereas the healthy mother expressed the wild-type G allele (Fig 2D). These results indicate that *CADPS2* is maternally expressed in blood. As mentioned earlier, in the 2 siblings with the maternal *CADPS2* deletion, no *CADPS2* mRNA could be detected in the blood, in agreement with the lack of expression of the paternal copy of the gene.

**CADPS2 allelic expression analysis in brain**

Different human brain regions were analyzed to study *CADPS2* expression. For three adult controls, heterozygous at SNP rs2251761, RNA from amygdala, cerebellum, cerebral cortex, and entorhinal cortex was available. RT–PCR analysis showed high expression level in cerebellum and cerebral cortex and a lower

expression in amygdala and entorhinal cortex (Fig 2E, upper panel). Sequencing of the PCR products showed that *CADPS2* was always monoallelically expressed in the amygdala, whereas it was biallelic in the other areas (Fig 2F). *CADPS2* expression analysis in blood cDNA of the same individuals confirmed the monoallelic expression and indicated that the same allele was expressed in both amygdala and blood. These results suggested that *CADPS2* might be subjected to tissue-specific monoallelic expression and putative imprinting.

**Quantitative methylation analysis of CADPS2 CpG regions**

We undertook a quantitative epigenetic analysis of *CADPS2* CpG regions located in the promoter and first intron. The EpiDesigner BETA software was used to predict the CpG islands: Four amplicons including 92 CpGs were selected (Fig 3A); 63 of these CpGs were suitable for analysis by gene-specific amplification using *in vitro* transcription coupled with mass spectrometry (MS) (Supplementary Table S2). Analysis was performed on bisulfite-treated DNA

**Figure 2. *CADPS2* allelic expression in different human tissues.**

A  Allelic expression in blood cDNA of SNP rs2251761 (A/G alleles) and rs2074589 (A/C alleles) in two heterozygous individuals for the two SNPs: upper panel: sequence electropherograms from genomic DNA (gDNA) showing the heterozygous state, lower panel: sequence electropherograms from blood cDNA; the SNP position is shown by the underscore marking.

B  Allelic expression from blood cDNA of SNP rs2251761 in an informative control family: the expressed A allele in the offspring inherits it from the mother.

C  Allelic expression of the variant p. 402Ala=: the change is inherited from the heterozygous father, who actually expresses only the variant allele in blood, whereas the affected child expresses only the maternal allele in blood.

D  Allelic expression of the variant p. Val1137Met: The maternally inherited variant is the only one expressed in blood cDNA of the affected son, whereas the healthy mother expresses the wild-type allele. The healthy brother does not inherit the change.

E  RT-PCR of *CADPS2* expression in different brain areas of one control individual heterozygous at SNP rs2251761; upper panel: PCR products covering *CADPS2*x1-x4 (encompassing SNP rs2251761); lower panel: PCR products of housekeeping gene *GUSBP*. Rt- = no reverse transcriptase in reaction; Rt+ = reverse transcriptase added.

F  Electropherograms showing *CADPS2* allelic expression in different brain areas of one control individual heterozygous at SNP rs2251761. In the lower panel, it is shown that the allele expressed by amygdala and blood cDNA is the same. gDNA = genomic DNA.

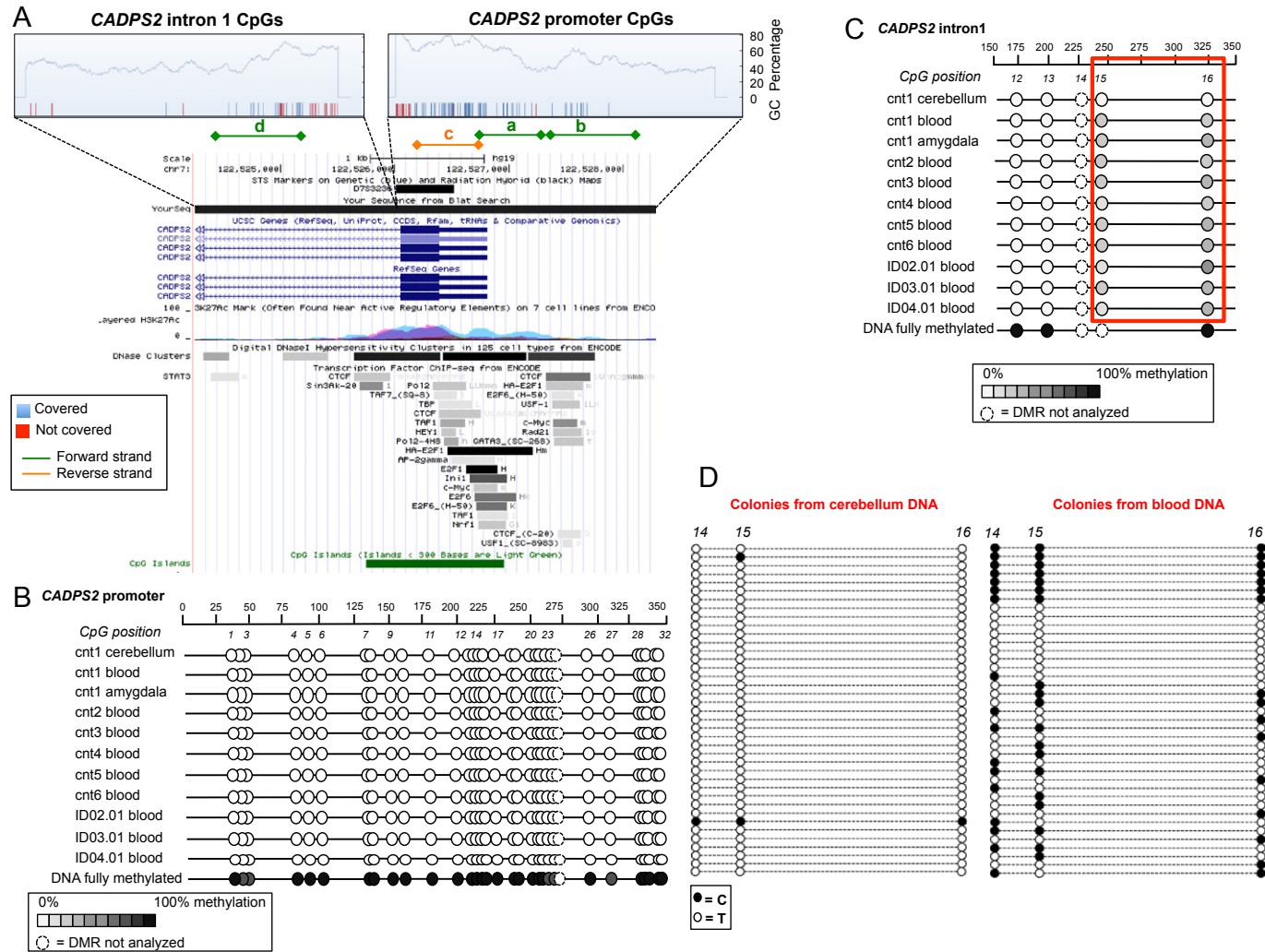

**Figure 3.  Methylation analysis of *CADPS2* promoter and intron 1 CpG regions.**

A    Upper panel: Epidesigner beta output of CpG prediction in the promoter and intron 1 of *CADPS2* genomic region; blue bars = CpG covered by MS analysis, red bars = CpG that cannot be analyzed by MS analysis, (a,b,d) sequences covered in the MS analysis using a forward primer in the *in vitro* transcription, (c) sequence covered by MS analysis using a reverse primer in the *in vitro* transcription (see text for details); lower panel: transcription factors, CpG island prediction, and regulatory sites in the corresponding genomic region as reported in UCSC Genome Browser (hg19).

B, C  Epityper beta output of *CADPS2* promoter and intron 1 quantitative methylation analysis via MS in cerebellum, amygdala, and blood genomic DNA bisulfite-treated from different individuals; the percentage of differential methylation is indicated by the different shades of gray as shown in the corresponding box; numbers indicate the base pairs relative to the amplified PCR product (upper lane) and the position of the CpG (lower lane); the differential methylation pattern is appreciable for CpG_15 and CpG_16 in intron 1.

D    Colonies carrying either the T (unmethylated, white circle) or the C (methylated, black circle) alleles at CpG_14, CpG_15, and CpG_16 of *CADPS2* from the colonies obtained by cloning the intron 1 PCR products from bisulfite-treated DNA of cerebellum and blood of individual cnt1.

extracted from blood, on bisulfite-treated DNA extracted from cerebellum, amygdala, and blood of one control individual (cnt1; heterozygous for SNP rs2251761), from amygdala and cerebellum of a second control (cnt2; heterozygous for SNP rs2251761), and from amygdala of another control individual (cnt3). One region in *IGF2* gene was used as known imprinted gene (Izzi *et al*, 2012; Supplementary Table S2).

Preliminary analysis on 9 blood DNA (3 ID and 6 control individuals) showed that the promoter region was unmethylated across all the three regions analyzed (Fig 3A and B); an example of the pattern is reported in Fig 3B (amplicon c, Supplementary Table S2).

Conversely, we identified two differentially methylated sites in the first intron of *CADPS2*, corresponding to the genomic coordinates: chr7: 122,525,608 (CpG_15) and 122,525,525 (CpG_16). For these two sites, cerebellum DNA showed a complete unmethylated status (Fig 3C and Table 2), whereas in blood and amygdala a consistent hemimethylated pattern was observed. The differences in methylation between cerebellum and blood (cnt1) and between cerebellum and amygdala (cnt2) were significant for both sites (Table 2), also when comparing the data from cerebella and amygdala from distinct individuals (CpG15, *P*-value = 0.0012; CpG16, *P*-value = 0.0102; Table 2). No differences were observed in the

**Table 2.  The two *CADPS2* DMRs (differentially methylated regions) validated by the MassARRAY analysis**

| DMR ID | Gene | cnt1 | | | cnt2 | | cnt3 |
|--------|------|------------|-------|----------|------------|----------|----------|
| | | **Cerebellum** | **Blood** | **Amygdala** | **Cerebellum** | **Amygdala** | **Amygdala** |
| Intron1CpG_15 | *CADPS2* | 0.000 | 0.243 | 0.230 | 0 | 0.185 | 0.215 |
| | | $P < 0.0001$[a] | | | $P = 0.0007$ | | |
| | | $P = 0.0012$[b] | | | | | |
| Intron1CpG_16 | *CADPS2* | 0.047 | 0.273 | 0.390 | 0.025 | 0.265 | 0.300 |
| | | $P = 0.0001$[a] | | | $P = 0.0338$ | | |
| | | $P = 0.0102$[b] | | | | | |
| SQNM | *IGF2* | 0.240 | 0.253 | 0.460 | 0.280 | 0.340 | 0.360 |
| | | $P = 0.7244$ | | | $P = 0.0859$[b] | | |

[a]Student's *t*-test (unpaired).
[b]*P*-values calculated between all cerebella and amygdala data.

methylation pattern for the control region SQNM (corresponding to the known imprinted gene *IGF2*). These data were corroborated by cloning the PCR products of intron 1 obtained from bisulfite-treated blood and cerebellum DNA of the same individual (cnt1) and colony screening for the presence of either C (methylated) or T (unmethylated) alleles at the CpGs (Fig 3D). This approach led us to analyze also CpG_14 (position 122,525,624) that was not possible to study with MS. Out of 39 colonies from blood, we found 43.58% methylation at position 122,525,624 (CpG_14), 48.72% methylation at position 122,525,608 (CpG_15), and 41.03% methylation allele at position 122,525,525 (CpG_16). Instead, the percentage of colonies derived from the PCR products of bisulfite-treated cerebellum DNA carrying the methylated allele was 2.56% at bp122,525,624 (CpG_14), 5.13% at bp122,525,608 (CpG_15), and 2.56% at bp122,525,525 (CpG_16), confirming the extensive unmethylation observed by MS analysis (Fig 3C). Quantitative data from bisulfite-treated blood DNA from controls and ID cases (for a total of 34 individuals) confirmed the presence of a hemimethylated pattern for CpG15 and 16 (Supplementary Table S3). In order to detect whether these sites (CpG_14, CpG_15, CpG_16) show a parent-of-origin-specific differential methylation, we analyzed the methylation status of four control individuals heterozygous for an adjacent SNP (rs981321, g.122,525,329 G > A), by performing colony analysis of an intron 1 fragment containing SNP rs981321 and the three CpG sites. In one control individual, we could detect a statistically significant preferential methylation of rs981321-A allele at CpG_16 (Fig 4A and B), while CpG15 and CpG14 did not show a differential methylation pattern for the two alleles; however, it was not possible to determine the parental origin of the two alleles in this individual (Fig 4A and B). For the three other heterozygous controls, the parental origin of the alleles was known; however, we did not detect a difference in the methylation level of the maternal and paternal alleles at any of the three CpGs (Fig 4C).

**Cadps2 allelic expression analysis in mice**

*Cadps2* expression was also evaluated in murine tissues. FVB and C57B/6 mouse strains were sequenced for two *Cadps2* coding SNPs (Supplementary Table S3). At SNP rs33756726, FVB mice were homozygous for the C allele and C57B/6 were homozygous for the T allele (Fig 5A). We analyzed *Cadps2* expression in tissues derived from heterozygous offspring of C57B/6 (mother) × FVB (father) crosses at different developmental stages: E17.5 d.p.c. ($n = 5$), P5, ($n = 4$), P20 ($n = 2$) (Fig 5B). Cerebral cortex, kidney, and heart showed a biallelic expression of *Cadps2* mRNA at all stages, whereas a monoallelic maternal expression was observed in the embryonic cerebellum (E17.5). This monoallelic expression was not retained postnatally (P5) and in the adult stage (P20).

RNA from amygdala was available only for mouse adult stage P20 and *Cadps2* expression was biallelic, at difference with the data obtained from human tissues.

# Discussion

We provide evidence that *CADPS2* shows tissue-specific monoallelic expression, with the maternally inherited allele expressed in human blood and in specific brain regions (in the amygdala at least).

Several studies demonstrated that a region on human chromosome 7 contains a cluster of imprinted genes (Riesewijk *et al*, 1997; Kosaki *et al*, 2000; Lee *et al*, 2000; Schneider *et al*, 2012), but no data on *CADPS2* regulation have yet been reported. *CADPS2* is an excellent candidate for neurologic development abnormalities, given that it is predominantly expressed in the nervous system, is involved in neurotrophin-3 (NT-3) and brain-derived neurotrophic factor (BDNF) release, and *cadps2*-knockout mice show abnormalities in neuronal development and impairments in social behavior (Sadakata *et al*, 2004, 2006, 2007b; Sadakata & Furuichi, 2009, 2010). Nevertheless, comprehensive data on the role of CADPS2 in human disorders are still lacking. Some deletions in the region are reported (see Decipher, ISCA), but are often very large with no details on phenotype and parent of origin. To our knowledge, only one large *de novo* deletion of 5.4 Mb encompassing *CADPS2*, as many other genes, was thoroughly described in a 3-year-old child with an eye disorder (attributed to the deleted *TSPAN12*), ASD diagnosis, dysmorphic features, and occipital epileptic discharges (Okamoto *et al*, 2011). A duplication overlapping exon 1 of *CADPS2* was identified in a single individual out of 2,026 healthy children through SNP-microarray analysis (Shaikh *et al*, 2009; nsv 524192 in

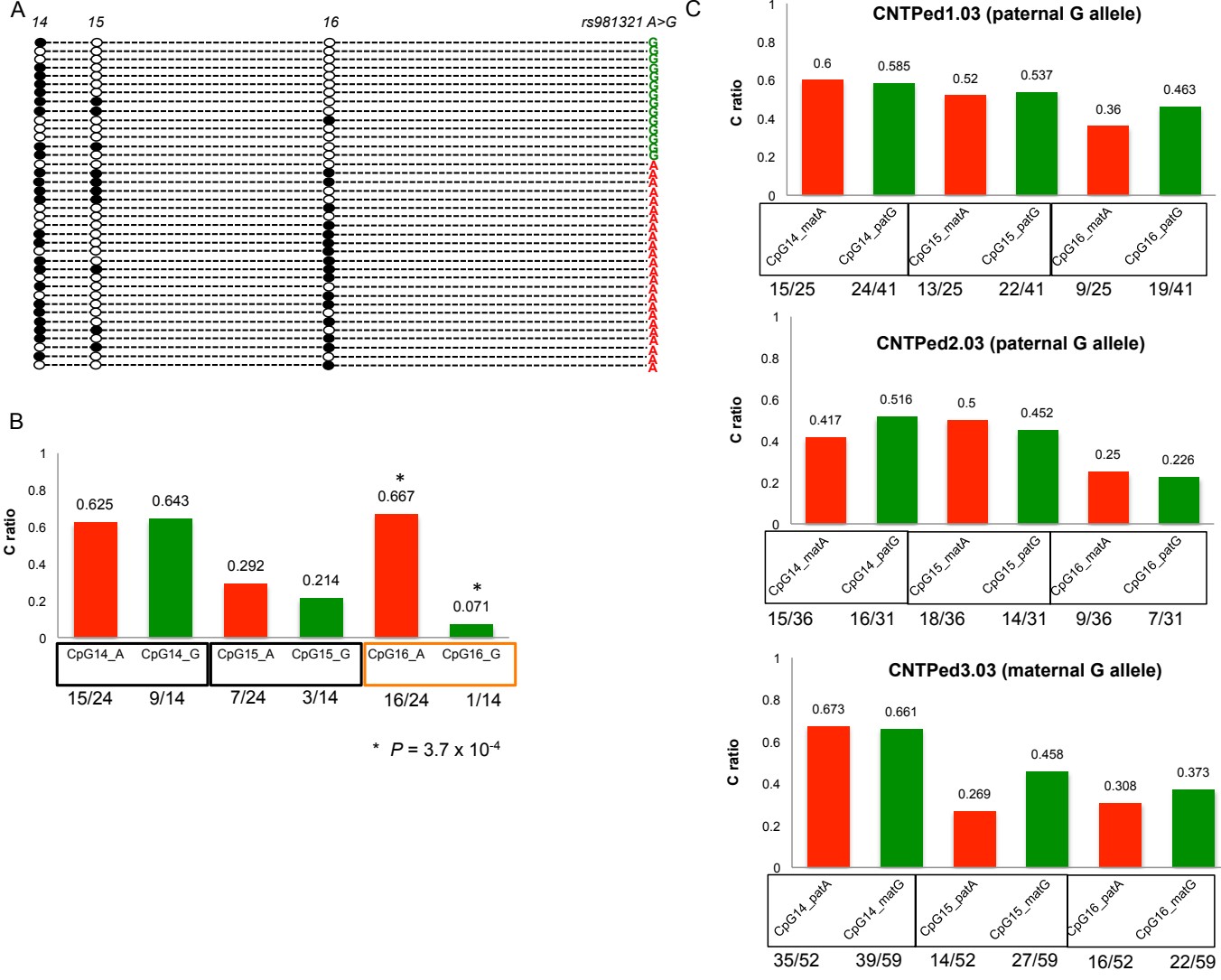

**Figure 4. Colony screening for parent-of-origin methylation analysis.**

A  Colonies carrying either the T (unmethylated, white circle) or the C (methylated, black circle) alleles at CpG_14, CpG_15, and CpG_16 and either G (green) or A (red) alleles at SNP rs981321 (A/G in *CADPS2* intron 1 PCR product; sequencing analysis performed on the colonies carrying the PCR product for intron 1 obtained from bisulfite-treated blood DNA of a heterozygous individual for SNP rs981321.

B  Histogram showing the ratio of colonies with the C methylated allele, for each CpG, for each rs981321 allele. Lower line: number of colonies carrying the C methylated allele out of the total number of colonies with the same allele at rs981321. Significant *P*-values are marked with star (Fisher's exact test).

C  Parental origin of the methylated and unmethylated alleles at CpG_14, CpG1_15, and CpG_16 for three individuals for whom parental origin of alleles at rs981321 was known as shown in the figure.

DGV); this CNV is of unknown functional significance and parental origin is not reported.

The relatively small intragenic deletion of *CADPS2* that we identified has not been reported before and we could not detect any *CADPS2* transcript in the blood of the two affected sibs carrying the deletion, suggesting that the maternal deleted allele is subjected to early mRNA decay and the undeleted paternal allele is not expressed.

Our findings point to a specific maternal expression of *CADPS2* in the blood, but also in the amygdala, that plays a critical role in social behavior (as part of the "social brain") and dysfunction of which has been implicated as a contributing factor in ASD (Schultz,

2005). The function of CADPS2 in amygdala has not yet been elucidated, nor its expression variation during development. However, recent published data have shown that human *CADPS2* expression is lower in the prenatal period and starts to increase in late fetal stage until mid-childhood in amygdala; the same trend is shown in cerebellum and neocortex (Kang *et al*, 2011).

A complex network involving amygdala and neurotrophin regulation emerges from studies in mice where changes in *Bdnf* methylation pattern were found during contextual fear learning (Suri *et al*, 2013), and from studies showing dysfunctions in amygdala and neurotrophin levels in neuropsychiatric disorders such as schizophrenia and depression (Nurjono & Chong, 2012; Kuhn *et al*,

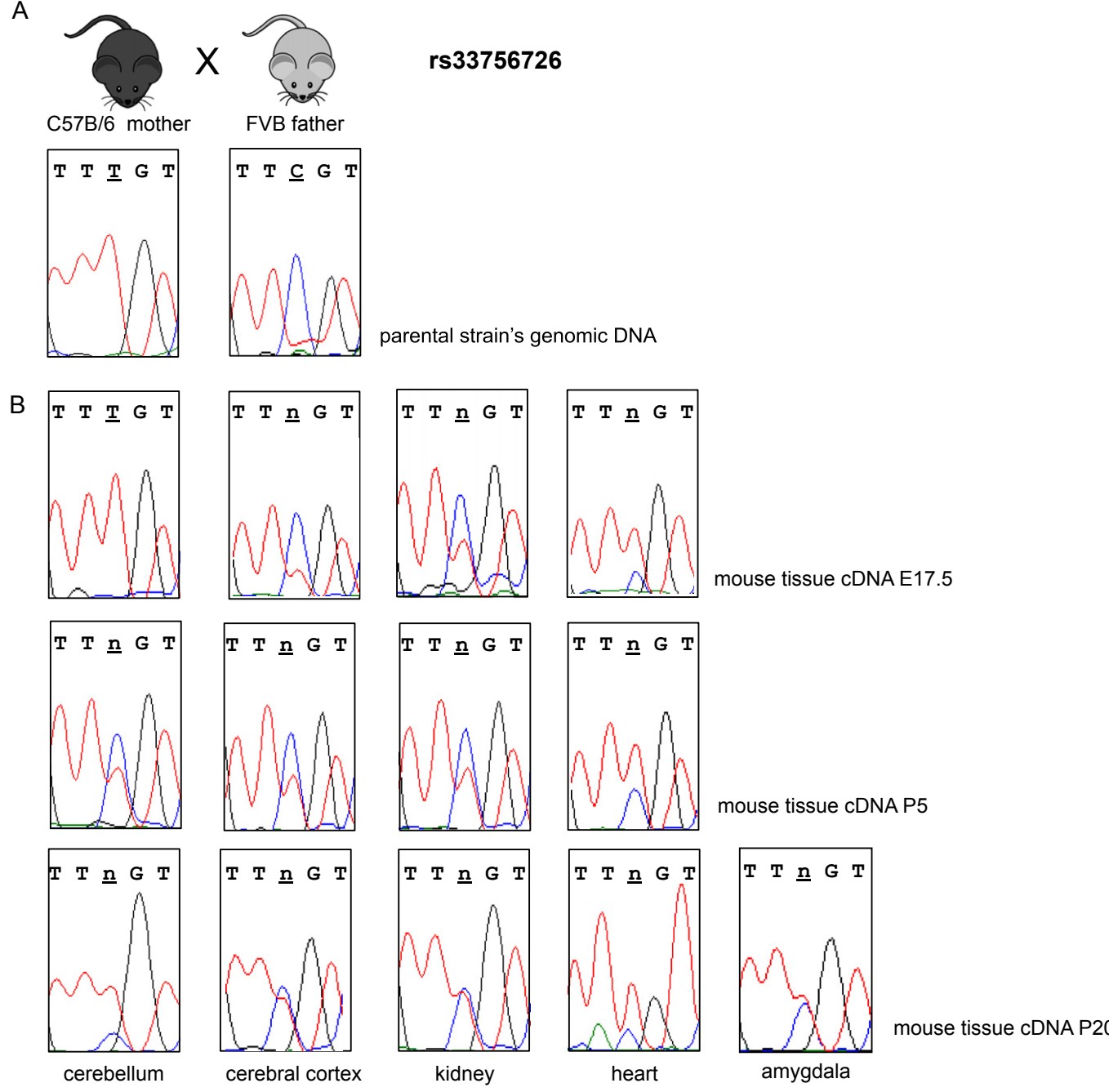

**Figure 5. Cadps2 expression in different mouse tissues from animals heterozygous for SNP rs33756726.**

A  Electropherograms of rs33756726 sequence in the parental genomic DNAs, showing that the mother C57/B6 is homozygous T and the father FVB is homozygous C.

B  Electropherograms showing rs33756726 allelic expression in different tissue cDNAs from the heterozygous offspring at different developmental stages (E17.5 d.p.c., P5, P20). A monoallelic expression is visible in the cerebellum at embryonic developmental stage E17.5.

2014) . Mutation in *CADPS2*, leading to its dysfunction, may alter these circuits.

We have identified maternally inherited putative damaging variants in ID/ASD cases. Sequencing of *CADPS2* had been previously performed on 90 autistic patients, but no disease-specific variants were identified (Cisternas *et al*, 2003), while eight heterozygous missense variants were identified in a cohort of 252 Japanese autistic

patients (Sadakata *et al*, 2007b); these variants were not investigated further, but it is worth noting that a p. Asp1112Asn was listed, which is adjacent and similar to one of the variants reported in the present paper. However, determining their effects is still a matter of question, since very few *CADPS2*-interacting proteins have been reported so far, and its exact role in neurotrophin release is not completely clear (Sadakata *et al*, 2007a, 2012a; Shinoda *et al*, 2011). Of note,

we could observe that one of the novel changes identified in this study, p. Asp1113Asn, disrupted CADPS2 binding to the dopamine receptor type 2, one of its few known interactors. However, its final effect on brain development and function is still far from being elucidated. Indeed, shaping human behavioral traits in animal models can be a challenge: The first study on *Cadps2* showed that only null homozygous mice had "autistic behaviors" and cerebellar defects (Sadakata & Furuichi, 2010), although later some "autistic traits" were also reported in heterozygous mice, even if the differences with wild-type animals were not statistically significant (Sadakata *et al*, 2013). Recently, behavior abnormalities were identified in homozygous mice expressing a *Cadps2* isoform lacking exon 3, previously associated with ASD and low IQ in humans (although these data are still controversial) (Sadakata *et al*, 2012b, 2013). Our data showed that *CADPS2* maternal expression in the adult amygdala is not conserved in mice; however, we identified a previously unreported monoallelic maternal expression in embryonal mouse cerebellum, suggesting that tissue- and temporal-specific monoallelic expression is present for *CADPS2* gene in mice and humans. Although we could not provide evidence for a parent-of-origin-specific methylation for *CADPS2*, further gene expression and methylation analyses of *CADPS2* genomic region, in diverse brain areas and at different developmental stages, are warranted in order to clearly elucidate the complex regulation pattern of this gene.

It is worth noting that from the comparison of the genome of prehistoric hominids, such as Neanderthals, with the present-day human genome *CADPS2* appears to have advanced during evolution to become a human trait-specific gene; the region containing *CADPS2* is one of the genomic intervals where positive selection in early modern humans was detected (Green *et al*, 2010). The additional regulation of *CADPS2* expression via monoallelic expression in the amygdala, which plays a key role in regulating social interactions, supports the importance of a fine modulation of *CADPS2* for human behavior.

Therefore, it is essential to reinterpret the available data on *CADPS2* variants (both sequence and copy number) gathered in other studies in light of parent-of-origin effect, in order to validate the role of *CADPS2* in neurodevelopmental disorders. Furthermore, large cohorts of trios of ID (Vissers *et al*, 2010) and ASD (Iossifov *et al*, 2012; Neale *et al*, 2012; O'Roak *et al*, 2012; Sanders *et al*, 2012) patients filtered for *de novo* variants (Veltman & Brunner, 2012) could be reassessed in order to verify whether additional maternally transmitted *CADPS2* variants are present.

We expect that regulatory mechanisms such as the one reported for *CADPS2* may contribute to the pathogenetic effect of variants and copy number variations in other genes inherited from otherwise healthy parents.

# Materials and Methods

### Patients and controls

#### Patients with CADPS2 deletion

The intragenic *CADPS2* deletion was identified in 2 siblings with similar clinical features, recruited in the framework of the FP-7 supported project CHERISH, which was approved by the local ethical review board. A 35-year-old man was followed for developmental delay and suffered a first generalized seizure at 5 years of age, when anti-epileptic therapy was initiated with good control. He was not self-sufficient and was reported to suffer frequent temper tantrums. Brain MRI and interictal EEG were normal. He did not have major birth defects, but an abdominal CT scan detected a supernumerary spleen. Currently, he has normal stature but is obese (height 178 cm, weight 111 kg, BMI 35). IQ evaluation (WAIS), performed at age 30, showed a borderline cognitive impairment. His sister, 33 years old, had seizures starting at age of 18 months, partially controlled by therapy, with an EEG showing paroxysmal parieto-temporal activity. IQ score (WAIS), at 28 years of age, was 76; brain MRI is normal. She is overweight (height 168 cm, weight 85 kg, BMI 30). They do not have other siblings.

#### ASD/ID individuals selected for CADPS2 mutation screening

ID patients were recruited in the framework of the project CHERISH. Before inclusion in the study, all patients underwent a detailed clinical ascertainment, routine molecular analysis to exclude fragile-X syndrome and metabolic disorders. 104 ID patients were recruited and were analyzed for CNV using the Agilent platform 44 K. 36 patients not showing any pathogenic rearrangement were included in *CADPS2* molecular mutation screening.

The patient carrying the pVal1137Met variant is a 22-year-old man. His family history is positive for kidney malformation (in a maternal aunt) and epilepsy (in the daughter of father's first cousin). The patient was born after an uneventful pregnancy but had perinatal distress, and presented preaxial polydactyly of his left hand. Psychomotor development was characterized by motor clumsiness and mild language delay. The first epileptic seizure occurred at 3 years and was characterized by left eye deviation, loss of contact, and hypertonia of the left side of the body. During follow-up, different seizures were observed: head nodding with upward deviation of the eyes; focal seizures with sudden fall. Seizures were difficult to control and their frequency was recently reduced with lamotrigine, clobazam, and levetiracetam. Clinical examination at 16 years showed macrocrania with occipital frontal circumference (OFC) of 60 cm, no major facial dysmorphisms, motor clumsiness, genu valgum, flat feet, obesity (height 168 cm, weight 102 kg, BMI 36), and mild intellectual disability (WISC-R). EEG recordings showed multifocal, diffuse, and generalized paroxysmal abnormalities. Brain MRI, fundus oculi examination, and abdominal ultrasound scan were normal.

The 94 Italian ASD individuals include 71 probands recruited at the Stella Maris Clinical Research Institute for Child and Adolescent Neuropsychiatry (Calambrone, Pisa, Italy), and 23 probands assessed at the Division of Child Neurology and Psychiatry, Department of Pediatrics, University of Catania. ASD diagnosis was based on the Autism Diagnostic Interview-Revised (ADI-R) and the Autism Diagnostic Observation Schedule (ADOS). A clinical evaluation was undertaken to exclude known medical disorders etiologically associated with autism (i.e., tuberous sclerosis, neurofibromatosis). Standard karyotyping, fragile-X testing, and EEG were obtained on all patients, and brain MRI was performed whenever possible. The male/female ratio of the affected individuals is 5.3:1.

Ninety-three additional unrelated probands with ASD were selected from 293 IMGSAC multiplex families. IMGSAC assessment methods and inclusion criteria used have been described previously (IMGSAC, 2001).

A blood sample was drawn from the propositus/a and available first-degree relatives.

The patient carrying the p. Asp1113Asn change was the first child of healthy non-consanguineous parents, born at term by spontaneous delivery after a pregnancy with no exposure or history of chronic illnesses, alcohol, tobacco, or street drugs. Family history was non-contributory. Birth weight was 3,200 g (50$^{th}$ centile), length 51.3 cm (50$^{th}$ centile), and OFC 33.8 cm (30$^{th}$ centile). Apgar was 10. She was breastfed for 2 months, thereafter fed with formula without any difficulties. Weaning to solid food was regular. She walked alone at 13 months. First words were reported around her first birthday. At 4 years, she was able to pronounce complete sentences with unintelligible speech verbal productions, pronominal inversions, and echolalia. She was withdrawn from early on, showing no interest toward her peers, being passive with inconstant eye-to-eye contact, with low level of frustration tolerance, and restricted interests. Play was very poor and repetitive. Gestual stereotypies were also observed. At kindergarten, she showed separation anxiety from her mother. Attention deficit and learning difficulties were reported in primary school. Sleep–wake rhythm was irregular, with frequent nocturnal awakenings. At age of 4 years, she started psychomotor and psychologic therapy and attended school with the help of a supporting teacher. ECG, EEG, brain MRI, complete metabolic work-up, and screening for celiac disease were normal. We first saw her at the age of 6 years. On physical examination, there were no dysmorphic features. Height was 123 cm (97$^{th}$ centile), weight 24 kg (90$^{th}$ centile), and OFC 51.5 cm (50$^{th}$ centile). Neurologic examination showed mild joint laxity and diffuse hypotonia. Language was characterized by the production of simple phrases with impaired prosody and, on occasion, echolalia. Comprehension was adequate whenever she paid attention to the verbal message. She had a mild cognitive impairment (WPPSI) without significant differences between verbal and performance competences. The evaluation for PDDs gave scores in keeping with a clinical diagnosis of PDD-NOS. Over time, she showed a slow, but constant, overall improvement.

*Human brain tissues*

Human brain samples from frontal cerebral cortex, amygdala, entorhinal cortex and cerebellum were obtained from deep-frozen (−80°C) slices of 3 adults showing either no significant histopathological changes (a 41-year-old woman) or neurodegenerative histopathological lesions of variable severity (a 78-year-old woman with mild AD pathology, and a 78-year-old man with dementia with Lewy bodies of neocortical subtype (Montine *et al*, 2012)).

All data from either patients or their caretakers and controls, including the informed consent, were handled in accordance with the local ethical committee's approved protocols and in compliance with the Helsinki declaration. Written informed consent for research use, given by the patients during life or by their next of kin after death, was also available for brain tissues used for RNA analyses.

### Array-CGH

Genomic DNA was extracted from peripheral blood and its quality was monitored using a NanoDrop spectrophotometer (Thermo Scientific, DE, USA). The entire array-CGH procedure performed according to the Agilent protocol for the 44K platform (Agilent Technologies, CA, USA). The new *CADPS2* deletion has been deposited in the ArrayExpress database (accession number: E-MTAB-2124).

### Microsatellite analysis

Two microsatellite markers mapping to *CADPS2* intron 1 (CADPS2_21xAC, genomic position: chr7:122,414,324-122,414,365) and intron 2 (CADPS2_26xAT, genomic position: chr7:122,348,575-122,348,626) were genotyped from blood-derived DNA of father, proband, and affected sister according to the following PCR conditions: 30 ng genomic DNA, 2.5 mM MgCl$_2$, 0.2 mM dNTPs, 0.5 μM primers (FAM-labeled), in a final volume of 10 μl using 0.25 unit of Gold Taq Polymerase (Life Technologies). A touch-down program of forty cycles was carried out as follows: 95°C 5′, 95°C 30″, 62–57°C, 72°C 30″ for 10 cycles, followed by 30 cycles at 95°C 30″, 57°C 30″, 72°C 30″ with a final extension of 7′ at 72°C. Samples were diluted 1/10 and 1 μl of dilution was run onto the automated 3730 ABI sequencing machine with LIZ(500) size marker (Life Technologies, Foster City, CA, USA). Genotype call was performed with GeneMapper v3.7.

### Real-time quantitative PCR

Quantitative PCR of *CADPS2* exons 3, 4, 6, 7, 26, 28, and 29 was carried out in blood-derived DNA from father, proband, affected sister, and one control sample, using Sybr-Green (Life Technologies). A region on *FOXP2* gene was amplified as internal control. PCR primers and conditions are available on request. Melting curve analysis of each PCR product was carried out to ensure specific amplification, and PCR efficiencies were calculated using a dilution series of DNA template. All samples were run in triplicate on the ABI7500 Fast PCR machine (Life Technologies). Relative copy number was calculated using the comparative Ct method, taking PCR efficiency into account.

### *CADPS2* breakpoint cloning

PCR primers for *CADPS2* deletion breakpoint cloning were designed with Primer3 v4.0 as follows: forward 5′-GGCAGAGAGGATGACG-TAG-3′; reverse 5′-CTGGATGGAGAAGAGCTGGA-3′. A long-range PCR was performed using the Expand-Long Range PCR kit according to the manufacturer's instruction (Roche Diagnostics) using 200 ng of genomic DNA from peripheral blood with the following PCR cycles (40): 92°C 2′, 92°C 10″ 60°C 15″ 68°C 14′, 68°C 7′.

PCR products were purified onto a Millipore PCR clean-up plate, cloned with the Original TA cloning kit (Life Technologies) in the pcDNA2.1 vector, and transformed into DH5α *E. coli* strains for white/blue screening. White colonies were grown and the plasmid DNA was purified and sequenced with universal M13 forward and reverse primers using the BigDye v1.1 kit (Life Technologies). Sequences were run onto the ABI 3730 automated sequencing machine, and electropherograms were analyzed with Chromas version 2.0.

### Mutation screening

PCR primers for human *CADPS2* (NM_017954.10) were designed with Primer3 v4.0. Genomic DNA extracted from peripheral blood

was amplified according to the following PCR conditions: 30 ng of DNA, 2.5 mM MgCl$_2$, 0.5 mM dNTPs, 0.5 µM primers, 5% DMSO in a final volume of 20 µl using the KAPA Fast Taq Polymerase Master mix (KAPA Biosystems, MA, USA). Forty cycles were carried out as follows: 95°C 1′, 95°C 15″, 58°C 15″, 72°C 20″, with a final extension of 30″ at 72°C. PCR products were purified onto Millipore PCR clean-up plates and directly sequenced on both strands using the BigDye v1.1 kit (Life Technologies). Electrophero-grams were visualized with Chromas version 2.0 and Sequencer version 4.7.

SNPs rs2074589 (chr7 g.122,078,414T > G) in exon 17 and rs20784589 (chr7 g.122,033,379A > G) in exon 22 were genotyped using the same PCR primers and conditions utilized for mutation screening.

### CADPS2 missense change analysis

Human *CADPS2* cDNA clone (NM_017954.10) was a kind gift of Dr. I. Eckhardt (Max-Planck-Institut für Experimentelle Medizin, Göttingen, Germany). The coding sequence was PCR-amplified and inserted in frame to the C-ter V5 tag in pcDNA3.1 vector using Pfx Taq Polymerase (Life Technologies) and site-directed mutagenesis was carried out using the QuickChange XL muta-genesis kit (Agilent Technologies) according to the manufacturer's instructions. All PCR conditions and primers are available on request. The insertion of the changes was verified by sequencing. $3 \times 10^5$ SHSY5Y human neuroblastoma cells (ATCC, UK) were plated for transfection of the different plasmids using liposomes according to the manufacturer's instructions (Lipofectamine, Life Technologies). Forty-eight hours after transfection, cells were lysed in 50 mM HEPES, 1 mM EDTA, 10% glycerol, 1% Triton X-100, 150 mM NaCl in the presence of protease inhibitors (Roche Diagnostics) and phosphatase inhibitors (Inhibition Cock-tail 2 and 3, Sigma). Pre-clearing was performed with rabbit IgG (Millipore) for 1 h at 4°C. Immunoprecipitation assays were performed at 4°C using 0.8 µg rabbit anti-D2DR antibody/reaction (Santa Cruz, CA, USA) or 1 µg mouse anti-V5 antibody/reaction (Life Technologies) on Protein G-Sepharose (Sigma). Proteins were separated by SDS gel electrophoresis, transferred onto nitro-cellulose membrane (GE Healthcare, UK), and subjected to western blotting with the Western Breeze kit (Life Technologies). Primary antibodies used were the following: mouse anti-V5 (Life Technologies) diluted at 1:5,000; mouse anti-tubulin gamma (Sigma) diluted at 1:10,000; rabbit anti-D2DR (Santa Cruz) diluted at 1:150. Bands were visualized by the ECL method (GE Healthcare).

### CADPS2 expression analysis

Total RNA from 1.5 ml fresh blood was extracted with the QIAGEN Blood Total RNA kit (QIAGEN, MD, USA). Total RNA from $5 \times 10^6$ lymphoblastoid cells, cultured as described (Bonora *et al*, 2002) or from human frozen brain tissues (30–40 mg) was extracted using with the QIAGEN Total RNA kit (QIAGEN). Tissues from F1 mice, obtained by crossing FVB (father) × C57B/6 (mother) lines, were obtained at different developmental stages from animals sacrificed by cervical dislocation. Kidney, heart, cere-bellum, cerebral cortex, and amygdala were collected for RNA

extraction and RT–PCR as described above. All animal procedures were carried out at San Raffaele Scientific Institute, Milan, Italy, according to, and approved by, the San Raffaele Institutional Animal Care and Use Committee.

One microgram of DNase I-treated RNA was used for reverse transcription with random hexamers using the Multiscribe RT system (Life Technologies) at 48°C for 40′ in a final volume of 50 µl. Six microliters of cDNA was used for testing *CADPS*2/*Cadps2* expression in human and mouse tissues, respectively. All primer pairs are available on request. PCR conditions were the following: 2.5 mM MgCl$_2$, 0.5 mM each dNTP, 0.5 µM primers, 5% DMSO in a final volume of 50 µl using the KAPA Fast Master mix (KAPA Biosystems). Forty-six cycles were carried out at 95°C 1′, 95°C 15″, 58°C 15″, 72°C 20″, 1′ at 72°C. PCR products were purified and sequenced as described before.

### MALDI-TOF MS methylation analysis

DNA methylation of *CADPS2* gene was determined by gene-specific amplification using *in vitro* transcription coupled with mass spec-trometry (MS) (MassARRAY platform, Sequenom, CA, USA; Ehrich *et al*, 2005; Di Vinci *et al*, 2012). One microgram of genomic DNA was submitted to bisulfite conversion, using the EZ DNA Methyla-tion Kit (Zymo Research, CA, USA), according to the manufac-turer's protocol, except for the conversion step consisting of 21 cycles at 95°C for 30 s and 50°C for 15 min. Universal unmethylated and methylated DNAs (Millipore, MA, USA) were used as internal controls. In addition, a 477-bp amplicon for the IGF2 region was included in the assay for quality assessment of bisulfite-treated DNA. A region length spanning 1.9 kb of promoter and 1.8 kb of intron 1 of *CADPS2* gene (corresponding to genomic coordinates chr7: 122,524,253–122,528,311, hg19) was submitted to EpiDesigner BETA software (Sequenom) for predicting the CpG islands. Four amplicons including 92 CpGs were selected; 63 of these CpGs were analyzable through this methodology (Supple-mentary Table S3).

The protocol employed a T7-promoter-tagged PCR amplification of bisulfite-converted DNA, followed by the generation of single-stranded RNA molecule and subsequent base-specific cleavage by RNase A. 25 µl H$_2$O MilliQ and 4 µg resin were added to the cleaved fragments. 20 nl of the analytes was nanodispensed onto 384-element silicon chips preloaded with matrix. Mass spectra were collected by using a matrix-assisted laser desorption/ionization time-of-flight (MALDI-TOF) MS, and spectra's methylation ratios were generated by EpiTYPER software V.1.2.22. Samples were run in duplicate/triplicate when possible. The different cleavage prod-ucts created from methylated or non-methylated target regions generate characteristic signal patterns that provide straightforward analysis by MALDI-TOF mass spectrometry. In analyzing the mass spectrum, the relative amount of methylation is calculated by comparing the difference in signal intensity between mass signals derived from methylated and non-methylated template DNA. The numerical value for each detected CpG unit/site is the detected methylation ratio found at that CpG unit/site after being treated with the bisulfite solution. Result data with blank cells are the result of the maximum level of uncertainty not being met. The maximum uncertainty level is entered as the uncertainty threshold. The recommended value for uncertainty threshold is 0.1, according

### The paper explained

#### Problem
Intellectual disability (ID) is a neurodevelopmental disorder characterized by a below-average score on tests of mental ability and limitations in daily life functions. Autism spectrum disorders (ASDs) are characterized by impaired social interactions and communication, stereotyped behaviors, and onset before 3 years of age. Both disorders are severe neuropsychiatric conditions, with overlapping clinical boundaries in many patients. Structural variants and single base pair mutations in genes involved in synaptic function have been identified in neurodevelopmental disorders; nevertheless, their pathogenic role is often unclear and causative mutations are still not found in a large proportion of patients.

#### Results
We identified a novel intragenic deletion in two siblings with borderline cognitive decline and epilepsy in *CADPS2* gene, encoding for a synaptic protein involved in neurotrophin release and interaction with dopamine receptor type 2 (D2DR).
Mutation screening in additional patients (187 with ASD and 36 with ID) identified several variants of maternal origin, including a missense change disrupting CADPS2/D2DR interaction. We showed that *CADPS2* was monoallelically expressed in blood and amygdala, and the expressed allele in the two tissues was the one of maternal origin. Few differentially methylated sites were identified in *CADPS2* first intron, in blood and amygdala, but they did not show a parent-of-origin methylation pattern typical of an imprinted gene. Nevertheless, we suggest that further gene expression and methylation analyses of *CADPS2* genomic region, in diverse brain areas and at different developmental stages, are warranted in order to clearly elucidate the complex regulation pattern of this gene.

#### Impact
We provide the first evidence that *CADPS2* shows a tissue-specific monoallelic expression, with only the maternally inherited allele expressed in human blood and in specific brain regions (in the amygdala to the least), and we indicate that *CADPS2* maternally inherited mutations can contribute to ID/ASD. We expect that regulatory mechanisms such as the one we discovered for *CADPS2* may contribute to the pathogenetic effect of variants and copy number variations inherited from otherwise healthy parents in other genes. In particular, we suggest that the available data on *CADPS2* variants (both sequence and copy number) gathered in other studies should be interpreted in light of parent-of-origin effect, in order to corroborate the role of *CADPS2* in neurodevelopmental disorders.

to the manufacturer's instructions, and any data with an estimated error larger than this value were excluded (a small value entered for this threshold causes the data displayed to exclude the more imprecise data and include more precise data for the selected amplicon). Sequenom peaks with reference intensity above 2, overlapping and duplicate units were excluded from the analysis (Izzi *et al*, 2012).

### Statistical analysis

Differences in frequencies between cases and controls were evaluated using the Fisher's exact test; quantitative differences in methylation ratio were evaluated using the Student's *t*-test (unpaired *t*-test); differences in the number of colonies carrying the methylated vs unmethylated alleles were calculated using the Fisher's exact test.

**Supplementary information** for this article is available online: http://embomolmed.embopress.org

### Acknowledgements
We thank all the patients, families, and healthy volunteers that participated in the study. We thank Prof. A. Contestabile for the support in mouse brain expression analysis and Ms. M. Giambartolomei and Dr. M. Vidone for technical help. This work was supported by FP7-EU Grant No 223692 "CHERISH" to G. R.

### Author contribution
EB, CG, EM, EB and MS designed the experimental plan, analyzed the data, and wrote the manuscript; FM and SL performed the mutation analysis and allelic expression; PM performed CNV analysis; CD, EM and VM performed quantitative methylation analysis; FB and MV performed the western blotting experiments; PP provided the human brain tissues; LR and MT provided the mouse tissues; AP, AB, LM, GT, the IMGSAC Consortium, and Giovanni Romeo provided ASD and ID samples and performed the clinical evaluation.

### Conflict of interest
The authors declare that they have no conflict of interest.

### For more information
http://www.cirp.org/library/ethics/helsinki
http://www.cherishproject.eu
http://dgv.tcag.ca/dgv/app/home
https://www.iscaconsortium.org/index.php
http://gvarianti.homelinux.net/gvariantib37/index.php
http://www.ebi.ac.uk/arrayexpress
http://rulai.cshl.edu
http://www.umd.be/HSF/
http://www.ncbi.nlm.nih.gov/SNP
http://www.1000genomes.org
http://evs.gs.washington.edu/EVS/
http://hbatlas.org/

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
