## [Review Process File · EMBO Molecular Medicine]

Maternally inherited genetic variants of CADPS2 are present in Autism Spectrum Disorders and Intellectual Disability patients

Elena Bonora, Claudio Graziano, Fiorella Minopoli, Elena Bacchelli, Pamela Magini, Chiara Diquigiovanni, Silvia Lomartire, Francesca Bianco, Manuela Vargiolu, Piero Parchi, Elena Marasco, Vilma Mantovani, Luca Rampoldi, Matteo Trudu, Antonia Parmeggiani, Agatino Battaglia, Luigi Mazzone, Giada Tortora, IMGSAC, Elena Maestrini, Marco Seri, Giovanni Romeo

Corresponding author: Marco Seri, University of Bologna

Review timeline:

Submission date:	27 June 2013
Editorial Decision:	10 September 2013
Revision received:	06 February 2014
Editorial Decision:	26 February 2014
Revision received:	08 March 2014
Accepted:	11 March 2014

Transaction Report:

Editor: Céline Carret

1st Editorial Decision

10 September 2013

Thank you for the submission of your manuscript to EMBO Molecular Medicine. I am sorry that it has taken so long to get back to you on your manuscript.

While reviewers 1 and 3 delivered their evaluations in a timely manner, we did not receive the other reviewers' input. As the evaluations from the first two reviewers are consistent, and a further delay cannot be justified, I have decided to proceed based on these evaluations.

As you will see, while referee 3 is rather supportive and suggests clarifications and further experimentation to strengthen the methylation data, referee 1 is much more critical and raises many important issues that will all have to be satisfactorily answered to render the data more compelling. As it stands, important details, numbers, and detailed analyses are missing, which make it difficult to properly evaluate the findings.

Given these evaluations, however, I would like to give you the opportunity to revise your manuscript, with the understanding that the referees' concerns must be fully addressed. Please note that it is EMBO Molecular Medicine policy to allow a single round of revision in order to avoid the delayed publication of research findings. Consequently, acceptance or rejection of the manuscript will depend on the completeness of your responses included in the next final version of the manuscript.

I look forward to seeing a revised form of your manuscript as soon as possible.

***** Reviewer's comments *****

Referee #1 (Remarks):

Bonora et al. investigated an interesting candidate gene for ASD/ID, CADPS2. Their manuscript stems from the identification of a novel deletion that overlaps multiple exons of the gene in two siblings with mild ID and epilepsy. The authors subsequently screened 130 additional patients with ASD or ID for sequence level changes and uncovered a handful of missense variants in this gene. They then argue that CADPS2 is monoallelically expressed in blood and amygdala through studies involving allelic expression analysis in adult tissues, methylation analysis, and allelic expression analysis in mice.

My overall impression of the paper is that the manuscript presents some novel observations that are publishable somewhere, but more work is required to provide the necessary support for their conclusions. The authors present an interesting family with a deletion overlapping a plausible candidate gene (based on previous reports in both humans and mice that suggest an association with ASD/ID). This is a novel discovery as, to our knowledge, no deletions or duplications have been found overlapping this gene in ASD/ID cohorts. The claim that the gene is maternally expressed is also novel (though I do have some issues with the way this was presented). The experiments performed are technically sound, but additional samples and/or further functional work is required to convince us of a genotype-phenotype correlation for the variant.

Specific comments:

The authors reported one intragenic deletion of CADPS2 but have not mentioned how many samples have been investigated. Also, are there any other CNVs found in the affected individuals? The authors mentioned it as a novel deletion, but they have not mentioned which CNV databases they have checked in (they should consider at least checking the Database for Genomic Variants). They should mention the duplication overlapping exon 1 from Shaikh 2009.

The authors have screened for mutations at CADPS2 in 130 ASD and/or ID individuals by Sanger sequencing, but this is a small sample size. Cisternas et al (2003 Genomics) have already sequenced 90 autism samples 10 years ago and found no significantly associated variant. The authors should cite this paper and discuss. In fact, when considering all the variants the authors detected in this study, only one (p.Asp1113Asn) seems to be potentially damaging. All the other missense variants are present in both the exome variant server and dbSNP (the statement in the paper that the variants are absent from other databases on page 5 is false) and should be clearly stated on pages 4 and 5. Because of this, the parental inheritance of these families is basically irrelevant. Including the deletion, there are only two clinically relevant families reported in this study (the family with the deletion and the family with p.Asp1113Asn). Although both are inherited from the mother, it is hard to conclude the maternal effect based on two cases.

For the allelic expression study, the problem again is the sample size. Although it is quite convincing from their experiments that the gene is expressed monoallelically, it is not compelling that it is an imprinted gene. It is now well-known that monoallelic expression is common in human genes (Gimelbrant et al 2007 Science). In fact, CADPS2 was listed as one of the monoallelically expressed genes from the Gimelbrant et al paper. However, monoallelic expression is not equal to imprinting. The fact that the authors used one family's parental origin of expression to support its maternal expression is clearly not enough. Therefore, the word "imprinting" used throughout the manuscript should be avoided (or changed into "putative imprinting").

The RT-PCR experiment shows relatively higher expression in cerebellum and cerebral cortex which might reflect the fact that both parental copies are expressed. The monoallelic expression is observed in amygdala and entorhinal cortex. The authors claim that amygdala plays critical role in social behavior; hence, disruption of this gene might lead to social dysfunction. The concerning point here is that monoallelic expression in amygdala is relatively low and it is not clear how disruption will lead to a phenotype. The analysis should include monoallelic expression pattern/methylation analysis in different age groups (i.e. early/late prenatal, childhood, adult) to detect an expression peak. The monoallelic expression is also not conserved in mouse, though the

gene exons are highly conserved. In their mice data, the most skewed maternal allelic expression is shown in cerebellum. It would be interesting to test if the methylation status is different in prenatal, postnatal and adult mice. The author should discuss and highlight these issues in the discussion for future analysis. Also, CADPS2 knockout mouse have impaired cerebellar development (Sadakata et al. 2007).

For the methylation analysis, the authors only investigated the allelic methylation without checking the parent-of-origin of the methylated allele. This is unfortunate, because the results may potentially support the putative imprinting status. They also haven't mentioned the total number of samples studied. Eventually, they only showed allele-specific methylation of one sample, which is again, is not enough to support for the imprinting claim.

Authors should be careful to account for the possibility of germline mosaicism rather than assume maternal inheritance of the deletion throughout the paper as the mother's DNA was not available for testing. They account for this in one section of the paper, but their statements suggest that the deletion was of maternal inheritance thereafter.

In multiple sections of the paper, CADPS2 is incorrectly spelled CADSP2.

Minor comments:

Table 1: Use of more than one prediction algorithm is recommended, instead of using just only one (i.e Polyphen-2).

On page 2, in the abstract add the word "the" before CADPS2 on line 4 and 13.

On page 2, delete the word "a" in the phrase "confirming the presence of a tissue-specific imprinting regulation.

On page 3, first line of first full paragraph: specify what "often" refers to; in the last line delineate to what "large proportion" refers to.

On page 3, final paragraph: Replace second word "structural" with CNV.

On page 4: Watch capitalization in "Long-Range PCR"

On page 4: Is the deletion a clean break or were any bases added?

On page 4, last paragraph: Give dbSNP id and frequency in dbSNP and EVS for each of the missense mutations.

On page 5: The statement "absent in other databases" is false; all missense changes besides p.Asp113Asn are in dbSNP.

On page 5: last sentence of first full paragraph: Add an "s" to database.

On page 5, authors state that SHSY5Y cell line is immortalized dopaminergic neurons, but they should note that this isn't exactly true. The cells originate from a neuroblastoma, but express dopaminergic markers.

The result section shows variants with paternal inheritance too. Please rephrase the sentence "The structural and most single nucleotide variants found in CADPS2 are of maternal origin".

On Page 8: In the methylation analysis section of the Results, all "demethylated" should be changed into "unmethylated".

On Page 8: line 8, first word. It should be "hemimethylated" instead of "emimethylated".

On Page 8: The presentation of C or T epiallele is too confusing. Basically after bisulfite conversion, all unmethylated C will be converted to T while methylated C will remain unchanged. So instead of using C or T epiallele, the authors should simply use "methylated allele" and "unmethylated allele" to represent "C" and "T", respectively.

On page 9: In the discussion, replace "in the amygdala to the least" with "in the amygdala at least"

On page 10, line 5: Replace "were" with "have been"

On page 10: Replace "comprehensive data on CADPS2 role in" with "comprehensive data on the role of CADPS2 in"

On page 11: Why is the Kuhn reference bolded?

On page 12: Cite Iossifov et al. 2012 with Neale, O'Roak, and Sanders.

On page 14: Remove comma after CHERISH

On page 14, fourth line: add the word "the" before Agilent

On Page 14: the proband has a family history of epilepsy. It would be good to state from which side of the family and who were affected.

On Page 20, last line: "SD equal to or greater than 0.10,": does it mean 0.10% or 0.10 (10%)? SD of 10% is a huge technical variation for a methylation analysis.

Referee #3 (Comments on Novelty/Model System):

The paper clearly associates CADPS2 mutations with ASD/ID phenotypes. Demonstration of CADPS2 imprinted expression and parent of origin-dependent phenotype is novel and relevant even for nonspecialists

Referee #3 (Remarks):

In this paper, Bonora et al. report results indicating that mutations in the CADPS2 gene are involved in the origin of Autism Spectrum Disorders and Intellectual Disability (ASD/ID) and provide evidences that this gene is maternally expressed likely due to genomic imprinting. The authors first identified a 285 kb deletion in the CADPS2 gene in two siblings affected by ASD/ID. Then, by screening about one hundred additional patients, they found two missense variants predicting to be damaging by bioinformatic analysis and demonstrated that one of them is actually interfering with type 2 dopamine receptor interaction. Since all the missense variants and likely the intragenic deletion were of maternal origin, the authors analysed the gamete of origin-dependent expression of this gene. They demonstrated that CADPS2 is maternally expressed in blood and monoallelically expressed in amygdala, while it is biallelically expressed in other brain areas. They then found that three CpGs in the first CADPS2 intron are partially methylated and differentially methylated between parental alleles supporting regulation by genomic imprinting. Finally, they investigated CADPS2 imprinting in the mouse and found that this gene is not imprinted in this species.

Although it was known that CADPS2 fell in an autism susceptibility locus and that the *Cadps2*-knockout mice showed Autistic-like phenotypes, no human mutation had been clearly associated with ASD/ID phenotypes so far. A few variants however were reported by Sadakata et al (2007) and not investigated further. Interestingly, one of the Sadakata's variants is adjacent and similar (Asp1112Asn) to the damaging Asp1113Asn identified in this study. The authors should comment on the possible functional significance of these additional variants.

CADPS2 map to a chromosome with multiple imprinted genes. The demonstration of tissue-specific maternal expression of CADPS2 is novel and very interesting. The methylation data, however, probably need a few confirmatory experiments. The authors find only three CpGs that are partially methylated and demonstrate a bias in allelic methylation in only one individual. This can also have causes other than imprinting. In my opinion, to support a functional role of these methylated CpGs, they should confirm the result in other individuals and ascertain the parental origin of methylation.

Minor points

- The absence of CADPS2 expression in the blood of the patient carrying the intragenic deletion is an important point and needs to be confirmed by Q-RTPCR and shown in a figure.

1st Revision - authors' response

06 February 2014

Referee #1 (Remarks):

Bonora et al. investigated an interesting candidate gene for ASD/ID, CADPS2. Their manuscript stems from the identification of a novel deletion that overlaps multiple exons of the gene in two siblings with mild ID and epilepsy. The authors subsequently screened 130 additional patients with ASD or ID for sequence level changes and uncovered a handful of missense variants in this gene. They then argue that CADPS2 is monoallelically expressed in blood and amygdala through studies involving allelic expression analysis in adult tissues, methylation analysis, and allelic expression analysis in mice.

My overall impression of the paper is that the manuscript presents some novel observations that are publishable somewhere, but more work is required to provide the necessary support for their conclusions. The authors present an interesting family with a deletion overlapping a plausible candidate gene (based on previous reports in both humans and mice that suggest an association

with ASD/ID). This is a novel discovery as, to our knowledge, no deletions or duplications have been found overlapping this gene in ASD/ID cohorts. The claim that the gene is maternally expressed is also novel (though I do have some issues with the way this was presented). The experiments performed are technically sound, but additional samples and/or further functional work is required to convince us of a genotype-phenotype correlation for the variant.

Specific comments:

The authors reported one intragenic deletion of CADPS2 but have not mentioned how many samples have been investigated. Also, are there any other CNVs found in the affected individuals? The authors mentioned it as a novel deletion, but they have not mentioned which CNV databases they have checked in (they should consider at least checking the Database for Genomic Variants). They should mention the duplication overlapping exon 1 from Shaikh 2009.

We have now reported the total number of ID patients analyzed in the framework of the CHERISH project (Methods, pages 15-16): 104 Italian ID cases were analyzed by Agilent 44k Array CGH: one patient carried the deletion in *CADPS2*. Two additional CNVs were present in the proband but they are already reported in DGV as non-pathogenic. These data have been inserted in Supplementary table 1.

The 36 ID patients included in the mutation screening did not show any pathogenetic CNV. Each identified CNV was checked against the following databases: DGV (Database for Genomic variants; <http://dgv.tcag.ca/dgv/app/home>), ISCA (<https://www.iscaconsortium.org/index.php>) and the Troina databases (database of human CNVs: <http://gvarianti.homelinux.net/gvariantib37/index.php>).

The *CADPS2* intragenic deletion was absent from all databases.

We added in the text (page 4 Results, and page 12, Discussion) the mentioned reference to the duplication overlapping exon 1.

The authors have screened for mutations at CADPS2 in 130 ASD and/or ID individuals by Sanger sequencing, but this is a small sample size.

The number of cases screened for mutation has been now increased to 223 with the inclusion of 94 unrelated individuals with ASD from multiplex families of the International Molecular Genetic Study of Autism Consortium (IMGSAC).

Cisternas et al (2003 Genomics) have already sequenced 90 autism samples 10 years ago and found no significantly associated variant. The authors should cite this paper and discuss.

This work has now been cited and discussed (page 13, Discussion).

In fact, when considering all the variants the authors detected in this study, only one (p.Asp1113Asn) seems to be potentially damaging. All the other missense variants are present in both the exome variant server and dbSNP (the statement in the paper that the variants are absent from other databases on page 5 is false) and should be clearly stated on pages 4 and 5. Because of this, the parental inheritance of these families is basically irrelevant. Including the deletion, there are only two clinically relevant families reported in this study (the family with the deletion and the family with p.Asp1113Asn). Although both are inherited from the mother, it is hard to conclude the maternal effect based on two cases.

Table 1 reporting coding variants identified through *CADPS2* screening has been amended: three variants were new, two of which are maternal. The p. Asp1113Asn is the only variant absent in controls and predicted to be damaging. This has been clarified in the text (page 5, Results). We agree that the evidence of a parent of origin-specific effect from transmission data is weak, however we note that in all three pedigrees with paternally transmitted non-synonymous variants, these were found in controls and were not shared by the affected sib-pairs.

For the allelic expression study, the problem again is the sample size. Although it is quite convincing from their experiments that the gene is expressed monoallelically, it is not compelling that it is an imprinted gene. It is now well-known that monoallelic expression is common in human genes (Gimelbrant et al 2007 Science). In fact, CADPS2 was listed as one of the monoallelically expressed genes from the Gimelbrant et al paper. However, monoallelic expression is not equal to imprinting. The fact that the authors used one family's parental origin of expression to support its maternal expression is clearly not enough. Therefore, the word "imprinting" used throughout the manuscript should be avoided (or changed into "putative imprinting").

We agree with the reviewer's comment on the fact that monoallelic expression is not equal to imprinting, therefore we amended the manuscript by referring only to "monoallelic expression" and only suggesting a putative imprinting effect.

However, we would like to point out that maternal monoallelic expression of *CADPS2* from blood cDNA has been demonstrated not in one, but in 3 different families and for different variants: in one family for SNP rs2251761 (Figure 2B), one family carrying the silent change p. Ala402= (Figure 2C) and in one family carrying the missense change p.Val1137Met (Figure 2D).

The RT-PCR experiment shows relatively higher expression in cerebellum and cerebral cortex, which might reflect the fact that both parental copies are expressed. The monoallelic expression is observed in amygdala and entorhinal cortex. The authors claim that amygdala plays critical role in social behavior; hence, disruption of this gene might lead to social dysfunction. The concerning point here is that monoallelic expression in amygdala is relatively low and it is not clear how disruption will lead to a phenotype.

In the entorhinal cortex *CADPS2* resulted expressed at low levels, but it showed a biallelic expression, suggesting that *CADPS2* monoallelic expression seems not to be correlated to the expression levels, but it may be related to other regulatory mechanisms.

Nevertheless, we fully agree with the reviewer's concern regarding the low expression in amygdala, but only further functional studies will shed light on the role of *CADPS2* in amygdala and how this might be related to social behavior.

The analysis should include monoallelic expression pattern/methylation analysis in different age groups (i.e. early/late prenatal, childhood, adult) to detect an expression peak. The monoallelic expression is also not conserved in mouse, though the gene exons are highly conserved. In their mice data, the most skewed maternal allelic expression is shown in cerebellum. It would be interesting to test if the methylation status is different in prenatal, postnatal and adult mice. The author should discuss and highlight these issues in the discussion for future analysis. Also, CADPS2 knockout mouse have impaired cerebellar development (Sadakata et al. 2007).

The suggestions of the reviewer has been taken into account in the Discussion, as further analysis on the expression and methylation status of human *CADPS2* at different developmental stages would be very valuable in order to understand its fine regulation. Unfortunately we did not have access to such tissues, but published data have shown that human *CADPS2* expression is low in the prenatal period and starts to increase in late fetal stage until mid-childhood in amygdala; the same trend is shown in cerebellum and neocortex (Kang et al, Nature 2011). These data have been added in the Discussion (pages 12-13).

We also increased the number of adult individuals analyzed for the intron 1 methylation status, by studying the bisulphite-treated blood DNA for a total of 34 individuals (16 ID cases and 18 controls). We also included data from additional individuals' brain areas (in total 2 cerebella and 3 amygdalae analyzed for methylation in intron 1). These analyses confirmed that the intron 1 in the cerebella is demethylated, whereas the amygdalae showed a comparable hemimethylation as in blood (Table 2).

Moreover, we were able to provide data on *Cadps2* expression in mice, at two different developmental stages (embryonal stage E17.5d.p.c. and P5), in addition to the P20 already presented. Expression studies were performed on tissues from 5 embryos and 4 P5 obtained from the F1 crosses of C57 female x FVB males (Figure 4B).

For the methylation analysis, the authors only investigated the allelic methylation without checking the parent-of-origin of the methylated allele. This is unfortunate, because the results may potentially support the putative imprinting status. They also haven't mentioned the total number of samples studied. Eventually, they only showed allele-specific methylation of one sample, which is again, is not enough to support for the imprinting claim.

Additional experiments were carried out in order to detect a parent-of-origin specific methylation at the three CpG differentially methylated in intron 1, in three additional individuals for whom the parental origin of the alleles at SNP rs981321 was known. Unfortunately we could not confirm a parental-specific methylation for these three sites; these data have been now added in the Results, Discussion and presented in the novel Figure 4 (A-C) and the text has been amended accordingly.

Authors should be careful to account for the possibility of germline mosaicism rather than assume maternal inheritance of the deletion throughout the paper, as the mother's DNA was not available for testing. They account for this in one section of the paper, but their statements suggest that the deletion was of maternal inheritance thereafter.

Through the analysis of informative microsatellite markers and SNPs we were able to show that the two sibs carrying the *CADPS2* deletion inherited two different paternal chromosomes, ruling out the possibility of paternal germline mosaicism as an origin of the rearrangement. Therefore the deletion should be of maternal origin. The data are now displayed in Figure 1B and discussed in page 4 (Results).

In multiple sections of the paper, CADPS2 is incorrectly spelled CADSP2.

Minor comments:

Table 1: Use of more than one prediction algorithm is recommended, instead of using just only one (i.e Polyphen-2).

On page 2, in the abstract add the word "the" before CADPS2 on line 4 and 13.

On page 2, delete the word "a" in the phrase "confirming the presence of a tissue-specific imprinting regulation.

On page 3, first line of first full paragraph: specify what "often" refers to; in the last line delineate to what "large proportion" refers to.

On page 3, final paragraph: Replace second word "structural" with CNV.

On page 4: Watch capitalization in "Long-Range PCR"

On page 4: Is the deletion a clean break or were any bases added?

On page 4, last paragraph: Give dbSNP id and frequency in dbSNP and EVS for each of the missense mutations.

On page 5: The statement "absent in other databases" is false; all missense changes besides p.Asp1113Asn are in dbSNP.

On page 5: last sentence of first full paragraph: Add an "s" to database.

On page 5, authors state that SHSY5Y cell line is immortalized dopaminergic neurons, but they should note that this isn't exactly true. The cells originate from a neuroblastoma, but express dopaminergic markers.

The result section shows variants with paternal inheritance too. Please rephrase the sentence "The structural and most single nucleotide variants found in CADPS2 are of maternal origin".

On Page 8: In the methylation analysis section of the Results, all "demethylated" should be changed into "unmethylated".

On Page 8: line 8, first word. It should be "hemimethylated" instead of "emimethylated".

On Page 8: The presentation of C or T epiallele is too confusing. Basically after bisulfite conversion, all unmethylated C will be converted to T while methylated C will remain unchanged. So instead of using C or T epiallele, the authors should simply use "methylated allele" and

"unmethylated allele" to represent "C" and "T", respectively.

On page 9: In the discussion, replace "in the amygdala to the least" with "in the amygdala at least"

On page 10, line 5: Replace "were" with "have been"

On page 10: Replace "comprehensive data on CADPS2 role in" with "comprehensive data on the role of CADPS2 in"

On page 11: Why is the Kuhn reference bolded?

On page 12: Cite Iossifov et al. 2012 with Neale, O'Roak, and Sanders.

On page 14: Remove comma after CHERISH

On page 14, fourth line: add the word "the" before Agilent

On Page 14: the proband has a family history of epilepsy. It would be good to state from which side of the family and who were affected.

On Page 20, last line: "SD equal to or greater than 0.10,": does it mean 0.10% or 0.10 (10%)? SD of 10% is a huge technical variation for a methylation analysis.

The corresponding part has been amended, now in page 24 of Materials and Methods, with a more detailed description of the Uncertainty Threshold value (0.1), according to the protocol of MassARRAY EpiTYPER v1.2 Software User Guide-Sequenom.

All the comments have been addressed in the corresponding sections of the manuscript.

Referee #3 (Remarks):

In this paper, Bonora et al. report results indicating that mutations in the CADPS2 gene are involved in the origin of Autism Spectrum Disorders and Intellectual Disability (ASD/ID) and provide evidences that this gene is maternally expressed likely due to genomic imprinting. The authors first identified a 285 kb deletion in the CADPS2 gene in two siblings affected by ASD/ID. Then, by screening about one hundred additional patients, they found two missense variants predicting to be damaging by bioinformatic analysis and demonstrated that one of them is actually interfering with with type 2 dopamine receptor interaction. Since all the missense variants and likely the intragenic deletion were of maternal origin, the authors analysed the gamete of origin-dependent expression of this gene. They demonstrated that CADPS2 is maternally expressed in blood and monoallelically expressed in amygdala, while it is biallelically expressed in other brain areas. They then found that three CpGs in the first CADPS2 intron are partially methylated and differentially methylated between parental alleles supporting regulation by genomic imprinting. Finally, they investigated CADPS2 imprinting in the mouse and found that this gene is not imprinted in this species.

Although it was known that CADPS2 fell in an autism susceptibility locus and that the Cadps2-knockout mice showed Autistic-like phenotypes, no human mutation had been clearly associated with ASD/ID phenotypes so far. A few variants however were reported by Sadakata et al (2007) and not investigated further. Interestingly, one of the Sadakata's variants is adjacent and similar (Asp1112Asn) to the damaging Asp1113Asn identified in this study. The authors should comment on the possible functional significance of these additional variants.

We added the corresponding reference and comment in the text (page 13).

CADPS2 map to a chromosome with multiple imprinted genes. The demonstration of tissue-specific maternal expression of CADPS2 is novel and very interesting. The methylation data, however, probably need a few confirmatory experiments. The authors find only three CpGs that are partially methylated and demonstrate a bias in allelic methylation in only one individual. This can also have causes other than imprinting. In my opinion, to support a functional role of these methylated CpGs, they should confirm the result in other individuals and ascertain the parental origin of methylation.

See also reply to Reviewer 1:

We agree with the Reviewer's comment, therefore, we increased the number of adult individuals analyzed for the intron 1 methylation status, by studying the bisulphite-treated blood DNA for a total of 34 individuals (16 ID cases and 18 controls). We also included data from additional individuals' brain areas (in total 2 cerebella and 3 amygdalae analyzed for methylation in intron 1). These analyses confirmed that the intron 1 in the cerebella is demethylated and the amygdalae showed a comparable hemimethylation as in blood (Table 2). Additional experiments were carried out in order to detect a parent-of-origin specific methylation at the three CpG differentially methylated in intron 1, in three additional individuals for whom the parental origin of the alleles at SNP rs981321 was known. Unfortunately we could not confirm a parental-specific methylation for these three sites; these data have been now added in the Results, Discussion and presented in the novel Figure 4 (A-C) and the text has been amended accordingly. Therefore, we suggested that that further methylation analysis of the entire *CADPS2* genomic region and gene expression in different tissues and at different developmental stages may be very valuable in the future in order to fully elucidate the complex regulation pattern of this gene.

Minor points

- *The absence of CADPS2 expression in the blood of the patient carrying the intragenic deletion is an important point and needs to be confirmed by Q-RT-PCR and shown in a figure.*

We have inserted as Supplementary Figure 1 the RT-PCR gel images, showing that the two ID sibs carrying the deletion do not present any *CADPS2* expression in blood cDNA.

2nd Editorial Decision

26 February 2014

Thank you for the submission of your revised manuscript to EMBO Molecular Medicine. We have now received the enclosed reports from the referees that were asked to re-assess it. As you will see the reviewers are now globally supportive and I am pleased to inform you that we will be able to accept your manuscript pending the following final amendments:

-I would like to ask you to take a special care in solving the issues mentioned by referee 1.

I look forward to reading a new revised version of your manuscript.

***** Reviewer's comments *****

Referee #1 (Remarks):

The paper is much better but there are still some changes indicated in the letter that do not appear in the main paper. It should not be the role of the reviewers to fact check more than once. The authors should check everything or how can we trust the data. Some references indicated in the letter do not appear in the paper (unless I am reading an old file).

Referee #3 (Remarks):

The revised manuscript is substantially improved and deserves publication

2nd Revision - authors' response

08 March 2014

We prepared an additional file of the manuscript body, showing as highlights all the changes that were inserted according to the reviewers' criticisms and suggestions, in order to facilitate the identification of all the modifications and of the newly added data (in respect to the first submitted version). In regards to Reviewer 1's criticisms, we deeply regret that, despite careful re-reading,

some references were not correctly inserted, as he/she wisely pointed out. We now edited the whole manuscript accordingly.

In addition, we report the detailed list of minor points that were corrected (not included in the previous cover letter, although the changes had been already made) and the current page of the manuscript in which they can be found.

Reviewer 1's comments

In multiple sections of the paper, CADPS2 is incorrectly spelled CADSP2.

We have corrected the misspelling of the gene acronym throughout the text.

Minor comments:

Table 1: Use of more than one prediction algorithm is recommended, instead of using just only one (i.e Polyphen-2).

We added SIFT as a second prediction algorithm and we inserted in table 1 the corresponding cut-off value for each variant and the predicted effect.

On page 2, in the abstract add the word "the" before CADPS2 on line 4 and 13.

Added as required

On page 2, delete the word "a" in the phrase "confirming the presence of a tissue-specific imprinting regulation."

Modified in "to tissue and temporal specific regulation"

On page 3, first line of first full paragraph: specify what "often" refers to; in the last line delineate to what "large proportion" refers to.

Modified in: "although definite epidemiological data are lacking, causative mutations remain unknown in the majority of ID/ASD patients".

On page 3, final paragraph: Replace second word "structural" with CNV.

On page 4: Watch capitalization in "Long-Range PCR"

Corrected in current page 5.

On page 4: Is the deletion a clean break or were any bases added?

As indicated in Figure 1E, 3 additional basepairs were present at the breakpoint.

On page 4, last paragraph: Give dbSNP id and frequency in dbSNP and EVS for each of the missense mutations.

These data have been inserted in Table 1.

On page 5: The statement "absent in other databases" is false; all missense changes besides p.Asp1113Asn are in dbSNP.

The description of CADPS2 rare variants has been amended throughout (pages 5-6, results section)

On page 5: last sentence of first full paragraph: Add an "s" to database.

On page 5, authors state that SHSY5Y cell line is immortalized dopaminergic neurons, but they should note that this isn't exactly true. The cells originate from a neuroblastoma, but express dopaminergic markers.

We have now inserted (Page 6) the modified description of SHSY5Y as follows: "derived from neuroblastoma and expressing dopaminergic markers".

The result section shows variants with paternal inheritance too. Please rephrase the sentence "The structural and most single nucleotide variants found in CADPS2 are of maternal origin".

We rephrased the sentence as follows: "We noticed that the majority of the missense variants that were absent in Italian controls and co-segregated with the phenotype were of maternal origin".

On Page 8: In the methylation analysis section of the Results, all "demethylated" should be changed into "unmethylated".

The term has been modified throughout the text as "unmethylated" (page 9).

On Page 8: line 8, first word. It should be "hemimethylated" instead of "emimethylated".

The typo has been corrected on page 9.

On Page 8: The presentation of C or T epiallele is too confusing. Basically after bisulfite conversion, all unmethylated C will be converted to T while methylated C will remain unchanged.

So instead of using C or T epiallele, the authors should simply use "methylated allele" and "unmethylated allele" to represent "C" and "T", respectively.

We modified the description of “C” and “T” alleles as methylated and unmethylated (Page 9), accordingly.

On page 9: In the discussion, replace "in the amygdala to the least" with "in the amygdala at least"

Modified as required (Page 11).

On page 10, line 5: Replace "were" with "have been"

Modified as required (Page 11)

On page 10: Replace "comprehensive data on CADPS2 role in" with "comprehensive data on the role of CADPS2 in"

Modified as required (Page 11)

On page 11: Why is the Kuhn reference bolded?

We apologize for the typing error; it has been amended

On page 12: Cite Iossifov et al. 2012 with Neale, O'Roak, and Sanders.

The citation has been inserted.

On page 14: Remove comma after CHERISH

Modified as required (Page 14).

On page 14, fourth line: add the word "the" before Agilent

Modified as required (Page 15).

On Page 14: the proband has a family history of epilepsy. It would be good to state from which side of the family and who were affected.

The information has been inserted (Page 15).

On Page 20, last line: "SD equal to or greater than 0.10,": does it mean 0.10% or 0.10 (10%)? SD of 10% is a huge technical variation for a methylation analysis.

The corresponding part has been amended, now on page 23 of Materials and Methods, with a more detailed description of the Uncertainty Threshold value (0.1), according to the protocol of MassARRAY EpiTYPER v1.2 Software User Guide-Sequenom.

On behalf of my Co-Authors, I do hope that this revised manuscript is now suitable for publication in EMBO Molecular Medicine.

Thank you again for your attention and collaboration.